# Demystifying Cost-Efficiency in LLM Serving over Heterogeneous GPUs

**Youhe Jiang** [* 1 2]  **Fangcheng Fu** [* 3]  **Xiaozhe Yao** [* 4]  **Guoliang He** [* 1]  **Xupeng Miao** [5]  **Ana Klimovic** [4]  **Bin Cui** [3]  **Binhang Yuan** [2]  **Eiko Yoneki** [1]

## Abstract

Recent advancements in Large Language Models (LLMs) have led to increasingly diverse requests, accompanied with varying resource (compute and memory) demands to serve them. However, this in turn degrades the cost-efficiency of LLM serving as common practices primarily rely on homogeneous GPU resources. In response to this problem, this work conducts a thorough study about serving LLMs over heterogeneous GPU resources on cloud platforms. The rationale is that different GPU types exhibit distinct compute and memory characteristics, aligning well with the divergent resource demands of diverse requests. Particularly, through comprehensive benchmarking, we discover that the cost-efficiency of LLM serving can be substantially optimized by meticulously determining GPU composition, deployment configurations, and workload assignments. Subsequently, we design a scheduling algorithm via mixed-integer linear programming, aiming at deducing the most cost-efficient serving plan under the constraints of price budget and real-time GPU availability. Remarkably, our approach effectively outperforms homogeneous and heterogeneous baselines under a wide array of scenarios, covering diverse workload traces, varying GPU availablilities, and multi-model serving. This casts new light on more accessible and efficient LLM serving over heterogeneous cloud resources.

---

[*]Equal contribution [1]Department of Computer Science, University of Cambridge, Cambridgeshire, UK [2]Department of Computer Science and Engineering, The Hong Kong University of Science and Technology, Hong Kong, China [3]Department of Computer Science, Peking University, Beijing, China [4]Department of Computer Science, ETH Zurich, Zürich, Switzerland [5]Department of Computer Science, Purdue University, West Lafayette, Indiana, US. Correspondence to: Binhang Yuan <biyuan@ust.hk>, Eiko Yoneki <eiko.yoneki@cl.cam.ac.uk>.

*Proceedings of the 42nd International Conference on Machine Learning*, Vancouver, Canada. PMLR 267, 2025. Copyright 2025 by the author(s).

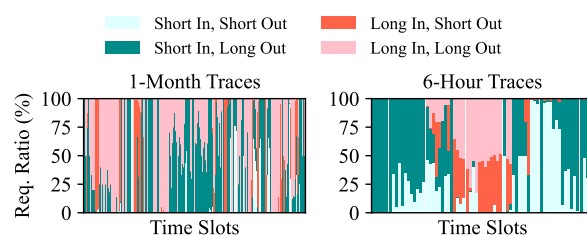

Figure 1: The real-world workload traces from the Swiss AI Center comprise over 500,000 traces collected over one month. We categorize the workload types based on input and output token lengths (longer than 512 and 128 are characterized as long).

## 1. Introduction

Large Language Models (LLMs), including GPT-4 (OpenAI, 2024), Gemini (Reid et al., 2024), Llama3 (Dubey et al., 2024), Claude (Anthropic, 2024), Mixtral (Jiang et al., 2024), and DeepSeek-V3 (Liu et al., 2024a), have demonstrated unprecedented performance across a wide range of real-world applications (GitHub, 2024; Jeon & Lee, 2023; Peng et al., 2023), such as chatbots, education, and healthcare, profoundly impacting human lives. In this context, enhancing the cost-efficiency of LLM serving is crucial for democratizing access to these cutting-edge technologies.

Currently, predominant practices utilize homogeneous GPU resources to deploy LLMs and serve the incoming requests (Li et al., 2023; Kwon et al., 2023; Agrawal et al., 2024b). However, with the broadening application domains, serving LLMs is facing progressively varying request patterns, driving the serving workloads dynamic and diverse—a phenomenon referred to as *workload heterogeneity* (Sun et al., 2024; Zhao et al., 2024b). This contradiction makes the use of homogeneous GPU resources unsuitable.

To be specific, the requests to be served have varying input and output token lengths, as exemplified by the real-world LLM serving traces at the Swiss AI Center shown in Figure 1. Such differences can exhibit significantly divergent resource (compute and memory) demands across different types of workloads, owing to the distinct characteristics of the two phases of inference— the prefill phase is compute-

Table 1: GPU Specifications and Pricing

| GPU Type | Peak FP16 FLOPS | Memory Access Bandwidth | Memory Limit | Price (per GPU) |
|---|---|---|---|---|
| A6000 | 91 TFLOPS | 960 GB/s | 48 GB | 0.83 $/h |
| A40 | 150 TFLOPS | 696 GB/s | 48 GB | 0.55 $/h |
| L40 | 181 TFLOPS | 864 GB/s | 48 GB | 0.83 $/h |
| A100 | 312 TFLOPS | 1555 GB/s | 80 GB | 1.75 $/h |
| H100 | 1979 TFLOPS | 3.35 TB/s | 80 GB | 2.99 $/h |
| 4090 | 83 TFLOPS | 1008 GB/s | 24 GB | 0.53 $/h |

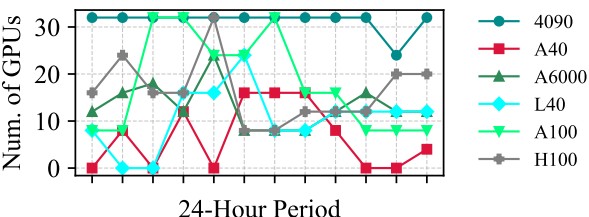

Figure 2: The number of different types of GPUs on Vast.ai during a 24-hour period.

bounded as it processes input prompts in a single step, while the decoding phase is memory-bounded as it generates subsequent tokens step by step (Zhong et al., 2024; Patel et al., 2024). Therefore, when using homogeneous GPU resources, it is hard to fit the varying resource demands well.

On the contrary, the heterogeneity in resource demands presents a unique opportunity to enhance the overall serving efficiency by leveraging different GPU types. As shown in Table 1, various GPU types offer diverse compute and memory capabilities, making them well-suited for processing different types of workloads. Motivated as such, we try to explore two questions: *Can serving LLMs over heterogeneous GPU resources achieve better cost-efficiency than homogeneous GPU resources? If yes, how can we enhance the cost-efficiency?* To this end, this work makes two technical contributions correspondingly.

***The first contribution*** is a comprehensive benchmarking of LLM serving over various GPU types, which offers a detailed understanding of cost-efficiency with heterogeneous GPU resources. Based on the benchmarking results, we reveal three key factors that are vital to the cost-efficiency:

- **GPU composition** (i.e., the number and types of GPUs that make up a heterogeneous cluster) is essential for optimizing the cost-efficiency of LLM serving. Different GPU types exhibit varying characteristics (e.g., computational capabilities, memory bandwidths, and memory capacities), making them more suitable for distinct workloads and model types. Given the varying types of incoming workloads, we need to strategically optimize GPU composition to improve resource utilization, reduce latency, and enhance overall serving performance.

- **Deployment configurations** (i.e., how many model replicas to deploy and the parallelism strategy for each) are necessary for maximizing overall system performance. The optimal configurations is influenced by the model, workload, and GPU type, so using a unique deployment configuration for all replicas is impractical. Therefore, we should adaptively optimize the deployment configurations so that the system efficiency can be improved.

- **Workload assignment** (i.e., allocating incoming workloads to GPUs) becomes crucial as different replicas are deployed with varying configurations (i.e., resources and

parallelisms) and different workloads have their preferable resource needs. As a result, to improve resource utilization, it necessitates assigning requests to more suitable replicas while balancing the burden across all replicas.

***The second contribution*** is to design a brand new LLM serving framework following the benchmarking, which aims at maximizing the cost-efficiency of LLM serving over heterogeneous GPU resources in cloud platforms.

Given the three factors above, a straightforward approach is to rent the most suitable GPUs for each workload type and assign requests accordingly. Nevertheless, this is impractical for two reasons. For one thing, due to the high demand for cloud GPUs, although cloud platforms (e.g., Vast.ai, RunPod, and AWS) offer a variety of GPU types, they usually have limited quantities of each type. We present the availability of different GPU types on Vast.ai over a 24-hour period in Figure 2. For another, the user-defined price budget is often constrained, making it impractical to always allocate sufficient GPUs for each workload demand.

Consequently, we formulate a scheduling algorithm based on mixed-integer linear programming (MILP), which takes account of both **real-time GPU availability** on cloud platforms and the **user-defined price budget**, while co-optimizing how to rent GPUs from the available pool (GPU composition), how to deploy the models over the rent GPUs (deployment configuration), and how to dispatch the workloads among the model replicas (workload assignment). We further incorporate practical heuristics and a binary search mechanism, as well as extend our approach to the multi-model scenario, improving scalability and solving efficiency for large-scale clusters.

We empirically evaluate our framework by comparing it with both homogeneous and heterogeneous baselines across a variety of scenarios, covering diverse workload traces, varying GPU availabilities, and multi-model serving. The results demonstrate that, within the same price budget, our approach can achieve up to 41% and on average 20% higher throughput, or reduce the serving latency by up to 54% and on average 20%.

## 2. Background

**LLM inference phase.** The inference process in LLMs consists of two main phases: the prefill phase and the decoding phase. During the prefill phase, the model processes the input prompt to compute the key-value (KV) cache and generates the first token in a single step. In contrast, the decoding phase uses the last generated token and the KV cache as inputs to generate subsequent tokens in a token-by-token manner. Generally, the prefill phase is compute-bound, while the decoding phase is memory-bound.

**Workload heterogeneity.** LLMs are designed to support a diverse range of applications, and the inference workloads associated with these applications often exhibit heterogeneity in input and output token lengths, which is called *workload heterogeneity*. Different workload types exhibit varying characteristics in terms of compute and memory demands. For example, requests from the WildGPT dataset (Zhao et al.), with average input and output token lengths of 496 and 510 respectively (classified as short input and long output), typically require more memory resources to handle the memory-bound decoding phase. Conversely, requests from the Azure-Trace dataset (Patel et al., 2024; Azure, 2024), with average input and output token lengths of 2455 and 18 respectively (classified as long input and short output), generally demand more compute resources to manage the compute-bound prefill phase. Therefore, appropriately allocating resources based on workload demands is critical for optimal performance.

**Heterogeneous LLM serving.** Recent research has explored various approaches for deploying LLM serving in heterogeneous GPU environments to achieve cost-efficient solutions (JIANG et al.; Miao et al., 2024; Griggs et al., 2024; Zhao et al., 2024a; Mei et al., 2024; Borzunov et al., 2023; Yan et al., 2024). HexGen introduces asymmetric partitioning and advanced scheduling techniques to deploy generative inference in decentralized and heterogeneous settings. Me'lange frames GPU allocation as a cost-aware bin-packing problem, optimizing cost efficiency for LLM services by effectively leveraging heterogeneous GPU types. Helix formulates the problem of heterogeneous GPU and network connection optimization as a max-flow problem, utilizing mixed-integer linear programming to determine the optimal model deployment. However, existing works typically optimize performance within a predefined heterogeneous cluster, and fail to consider GPU availability and user-defined budget constraints on cloud platforms. In addition, they are generally unaware of the workload heterogeneity, and only consider uniform workload assignment.

**LLM serving optimization.** There are several related works focusing on the optimization of LLM serving (Li et al., 2023; Yu et al., 2022; Kwon et al., 2023). QLM (Patke et al., 2024) focuses on SLO-aware serving and multi-node optimizations by refining request ordering; SarathiServe (Agrawal et al., 2024b) optimizes batching through prefill chunking to mitigate interference between the prefill and decoding stages; and Vidur (Agrawal et al., 2024a) develops an accurate simulator for deployment tuning. In contrast, our method is dedicated to achieving heterogeneous, cost-efficient serving in cloud environments.

## 3. Observation and Opportunity

Given a user with a specified budget estimation (in \$/h) renting GPUs from the cloud for serving certain workload traces, our objective is to deliver a comprehensive serving plan that maximizes the cost-efficiency of the user's serving system. In this section, we first benchmark the cost-efficiency performance of various workload types across different GPU types, model types, and deployment configurations. Then, we present our key observations and opportunities.

**Benchmark settings.** We subsample nine workload types from the ShareGPT (Zheng et al.), WildGPT (Zhao et al.), and Azure-Trace datasets (Patel et al., 2024). These workloads are characterized by average input token lengths of $\{2455, 824, 496\}$ and output token lengths of $\{510, 253, 18\}$. Each combination reflects distinct workload characteristics. For example, $\{2455, 18\}$ (long input, short output) represents compute-intensive workloads, while $\{496, 510\}$ (short input, long output) represents memory-intensive workloads. Based on these workload types, we evaluate two models, Llama3-8B and Llama3-70B, on six commonly used cloud GPUs (A6000, A40, L40, A100, H100, and 4090) with different deployment configurations. The benchmarking metrics include request throughput per unit cost (i.e., throughput divided by GPU cost) and the total cost associated with various latency percentiles (e.g., p5, p10, p15, ..., p95, p100). The total cost for each latency percentile is calculated by multiplying the latency time by the GPU cost. These metrics serve as indicators of cost efficiency. The GPU costs are demonstrated in Table 1.

**Observation-1: Heterogeneous GPUs are well-suited for managing model and workload diversity in LLM serving.** Figure 3 and Figure 11 present the benchmark results for the Llama3-70B and Llama3-8B models across various GPU types and workload types. The observations can be summarized as follows: **(i)** The H100 and A100 GPUs (data center GPUs) perform well on compute-intensive workloads with the Llama3-70B model, as both GPUs have high computational power to handle intense computational tasks. **(ii)** The A40, A6000, and L40 GPUs (workstation GPUs) excel in memory-intensive workloads with the Llama3-70B model. Workstation GPUs offer on average $1.2\times$ higher memory bandwidth and $1.8\times$ greater memory capacity per unit price than data center GPUs, making them more cost-efficient for memory-intensive workloads that often underutilize H100

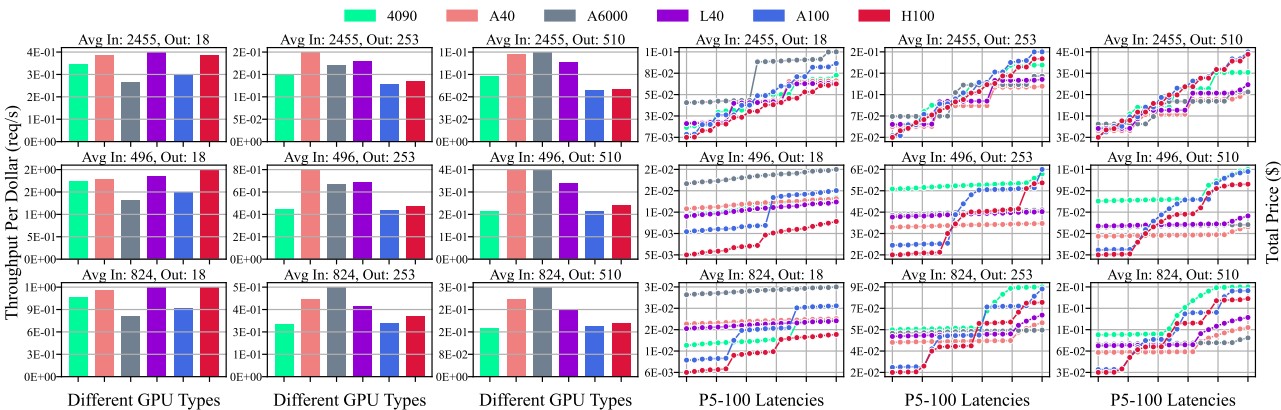

Figure 3: Benchmarked results for Llama3-70B model with different GPU types on different workload types. The left three columns represent the throughput results, x-axis represents different GPU types, y-axis represents throughput per unit price (i.e., throughput divided by GPU cost). The right three columns represent the latency results, x-axis represents the P5-100 latency results (P5-P100 Latencies means from left to right, the x sticks represent P5 Latency, P10 Latency, P15 Latency ...), y-axis represents total price (i.e., each latency time multiplied by GPU cost). Results for Llama3-8B model are demonstrated in Appendix A.

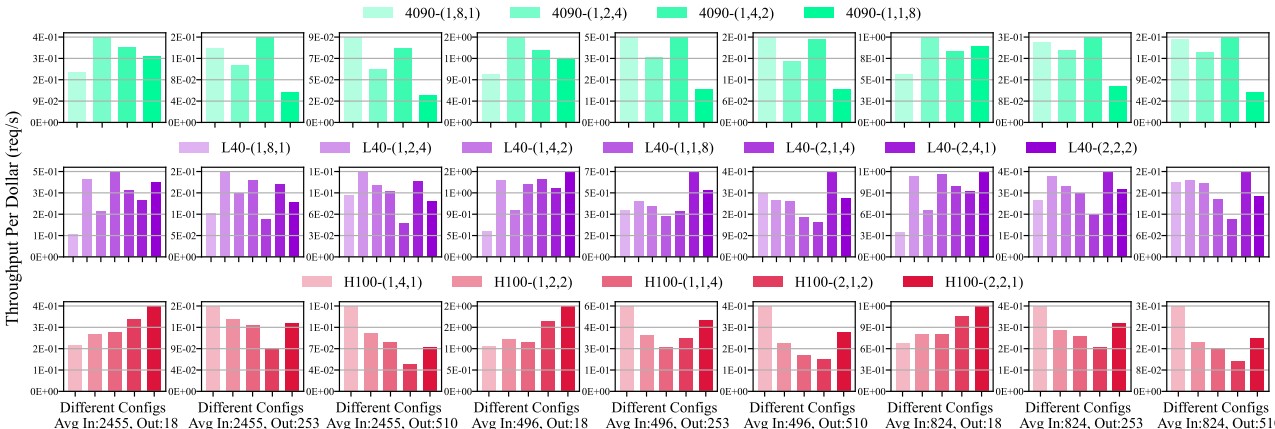

Figure 4: Throughput results for Llama3-70B model with different deployment configurations on different workloads. The three-element array represents the DP, TP, and PP degrees. Full benchmarking results are listed in Appendix B.

and A100 GPUs due to memory constraints. **(iii)** The 4090 GPUs (consumer GPUs) deliver excellent performance with the Llama3-8B model. As smaller models require significantly less compute and memory, and the consumer GPUs offer superior memory bandwidth per unit price, approximately $1.9\times$ higher than that of the A100 and H100 GPUs. Overall, our experimental results demonstrate that selecting the most appropriate GPU type for specific workloads and models can enhance the cost-efficiency performance of LLM serving by up to $2.27\times$. These findings underscore the necessity of aligning GPU types with model and workload demands to maximize both performance and cost-efficiency.

**Observation-2: Optimal deployment configurations are crucial for maximizing cost-efficiency across models,**

**workloads, and GPU types.** Figure 4 presents the benchmark results of various deployment configurations across different models, workloads, and GPU types. The observations can be summarized as follows: **(i)** Optimal configurations vary by workload type for a given GPU type. For instance, on H100 GPUs serving Llama3-70B, tensor parallelism (TP) (Shoeybi et al., 2019) is most effective for compute- and memory-intensive workloads (e.g., $\{2455, 510\}$), while higher degree of data parallelism (DP, i.e., model replication) performs better for less demanding workloads (e.g., $\{496, 18\}$). **(ii)** Optimal configurations vary by GPU type for a given workload type. For instance, in compute-intensive workloads (e.g., $\{2455, 18\}$), the L40 GPUs achieve the best performance using pure pipeline par-

allelism (PP) (Huang et al., 2019), while the H100 GPUs excel with a combination of DP and TP. (**iii**) Optimal configurations also depend on model type. For instance, with Llama3-8B models, DP consistently outperforms model parallelism (i.e., TP and PP). Since the Llama3-8B model has lower memory requirements and can run on a single GPU without model parallelism, increasing the number of model replicas (i.e., raising the DP degree) enhances the system's parallel processing capability, thereby improving cost efficiency. Overall, our experimental results demonstrate that selecting the most effective deployment configurations can improve system performance by up to 2.61×. These findings highlight the need for optimized deployment configurations to maximize cost-efficiency in LLM serving.

**Observation-3: The workload assignment should be co-optimized with the heterogeneous GPU composition and deployment configurations.** Effective workload assignment is critical for managing workload heterogeneity and achieving optimal performance. It serves two key objectives: (**i**) Directing requests to GPU types and deployment configurations that best match their resource demands to enhance serving efficiency. As noted in observation-1 and -2, even with an optimal GPU composition and deployment setup, performance may degrade if workloads are assigned to unsuitable GPUs or configurations. This misalignment can lead to inefficiencies and reduced overall performance. (**ii**) Balancing workloads across GPUs to prevent overloading or underutilization, thereby maximizing resource utilization. Workload balancing is essential for cost-efficient GPU utilization. In some cases, achieving full GPU capacity requires assigning workloads to suboptimal GPUs or configurations. While this may not be ideal for individual requests, it ultimately improves overall system performance.

**Constraints: Appropriate resource scheduling is crucial under limited resources and budget constraints.** Allocating workloads to the most suitable GPUs is a straightforward strategy for cost-efficient LLM serving. However, real-world deployments often face resource availability and budget limitations: (**i**) On cloud service platforms, GPU availability fluctuates, leading to shortages during peak periods. For instance, A40 availability on RunPod and Vast.ai can range from 3 to 16 and 0 to 32, depending on the time. (**ii**) Budget constraints may prevent users from selecting the optimal GPU for each workload, necessitating compromises in resource allocation. These constraints often force workload assignments to suboptimal GPUs, reducing performance and efficiency. To mitigate these challenges, an **effective scheduling algorithm** is essential. It should account for the user's budget constraints and real-time GPU availability, enabling efficient and adaptive LLM serving even under constrained conditions.

**Opportunities: Optimization of heterogeneous GPU de-**

**ployment for cost-efficient LLM serving.** Existing systems typically assume a *homogeneous GPU cluster* for LLM serving or focus on optimizing performance within a *predefined heterogeneous cluster*. However, adjusting the heterogeneous GPU composition within the serving cluster to align with specific workload demands offers a more cost-efficient alternative. Based on prior observations, we propose optimizing LLM serving by customizing the deployment of heterogeneous GPU types to meet workload requirements. This includes determining the optimal heterogeneous GPU composition (**observation-1**), selecting the most effective deployment configurations (**observation-2**), and implementing the most appropriate workload assignment (**observation-3**). Ultimately, our aim is to deliver a comprehensive LLM serving plan that meets user requirements, adapts to cloud environment constraints, and maximizes cost-efficiency (**constraints**).

## 4. Scheduling Algorithm

### 4.1. Problem Formulation

Given the LLMs to be served, a set of heterogeneous workloads, a user-defined budget, and GPU availability on the cloud, we seek a cost-efficient serving plan comprising: (1) **GPU composition**, i.e., selecting the type and number of GPUs to rent from the cloud while meeting budget and resource requirements; (2) **deployment configurations**, i.e., organizing the rented GPUs into serving groups, each responsible for serving one model replica, and determining their parallelism strategies; and (3) **workload assignment**, i.e., determining the allocation of incoming workloads across model replicas. Our objective is to minimize the overall makespan for processing all incoming workloads. The resulting plan must ensure that the user obtains the most **cost-efficient** LLM serving solution under the specified budgetary and resource constraints.

### 4.2. Simple Example

**Experiment setup.** We begin by assuming three GPU types, $\{t_1, t_2, t_3\}$, each with two units available. The hourly rental prices for these types are 4, 2, and 2 \$/h, respectively. We consider two workload types $\{w_1, w_2\}$, which arrive simultaneously with 80 total requests for $w_1$ ($\lambda_1 = 80$) and 20 total requests for $w_2$ ($\lambda_2 = 20$). We denote by $C_{t,w}$ the throughput (in requests per second) of GPU type $t$ on workload $w$. If each GPU serves one model replica, the throughputs are $C_{1,1} = 1.0$, $C_{1,2} = 1.2$, $C_{2,1} = 0.9$, $C_{2,2} = 0.9$, $C_{3,1} = 0.3$, and $C_{3,2} = 0.5$. Note that $C_{\sim,1}$ and $C_{\sim,2}$ vary with model parallelism. In **Cases 1** and **2**, we assume the workload is assigned to each GPU in proportion to its processing rate, so the system-wide throughput for each workload is the sum of individual-GPU rates. In **Case 3**, we allow workload-aware assignment for further optimization.

**Case 1: GPU composition.** We compare two compositions under the same budget of 8 \$/h, where each GPU serves one model replica. Composition 1 consists of $1 \times t_1$, $1 \times t_2$, and $1 \times t_3$, achieving a total throughput of $(2.2, 2.6)$ rps on $(w_1, w_2)$, with a processing time of $44.05$ s. Composition 2 consists of $1 \times t_1$ and $2 \times t_2$, achieving throughputs of $(2.8, 3.0)$ rps on $(w_1, w_2)$, with a processing time of $35.24$ s. Thus, modifying the GPU composition within the same budget results in a 20% speedup.

**Case 2: Deployment configuration.** Focusing on composition 2, we compare two ways to organize these three GPUs. Configuration 1 assigns each GPU to serve a single model replica, resulting in a processing time of $35.24$ s. Configuration 2 applies TP to the two $t_2$ GPUs, changing their combined throughput to $2.4$ rps on $w_1$ and $1.5$ rps on $w_2$, reducing the overall processing time to $30.94$ s. Thus, modifying the deployment configuration improves the processing time by approximately 14%.

**Case 3: Workload assignment.** Finally, we retain the same composition and TP-based configuration but optimize workload assignment. Specifically, we assign 15% of $w_1$ and 100% of $w_2$ to the replica with $t_1$, and 85% of $w_1$ to the replica with TP on $2 \times t_2$. With this assignment, the overall completion time is reduced to $28.67$ s. Thus, adjusting the workload assignment results in an additional 8% reduction in processing time.

This step-by-step example demonstrates the joint optimization of GPU composition, deployment configuration, and workload assignment for optimal performance. A detailed processing time calculation for each case is in Appendix C.

### 4.3. MILP Formulation

In this section, we introduce a mixed-integer linear programming (MILP) formulation to find a serving plan, i.e., GPU composition, deployment configurations and workload assignment, that minimizes the overall processing time. An overview of the symbols is shown in Table 2.

Let there be $N$ types of GPUs, indexed by $n \in \{1, 2, \ldots, N\}$. We denote the decision on how many GPUs of each type to use (i.e., **GPU composition**) by a vector $\mathbf{D} = [d_1, d_2, \ldots, d_N]$ where each $d_n \geq 0$ represents the number of GPUs of the $n$-th type. These variables are subject to availability constraints encoded by a vector $\mathbf{A} = [a_1, a_2, \ldots, a_N]$, such that $0 \leq d_n \leq a_n, \forall n = 1, 2, \ldots, N$. Each GPU type $n$ has price $p_n$ (e.g., 1.75 \$/h for A100, 2.99 \$/h for H100), and memory limit $m_n$ (e.g., 48 GB for L40, 80 GB for H100).

**Configurations.** We consider a set $\mathcal{C}$ of feasible configurations (i.e., **deployment configurations**). Each configuration $c \in \mathcal{C}$ represents the serving plan for a single model replica, which is characterized by $(v_c, s_c, o_c, h_{c,w})$:

Table 2: Symbols used in MILP.

| Symbol | Description |
|---|---|
| $N$ | number of GPU types |
| $W$ | number of workload types |
| $\mathcal{C}$ | set of feasible configs |
| $d_n$ | type $n$ GPUs allocated |
| $a_n$ | maximum available type $n$ GPUs |
| $p_n$ | rental price of type $n$ GPUs |
| $m_n$ | memory limit of type $n$ GPUs |
| $B$ | user-defined total price budget |
| $v_c$ | GPU composition of config $c$ |
| $s_c$ | parallelism strategy of config $c$ |
| $o_c$ | price cost of config $c$ |
| $h_{c,w}$ | throughput of config $c$ on workload $w$ |
| $x_{c,w}$ | assignment of workload $w$ to config $c$ |
| $y_c$ | whether config $c$ is used |
| $T$ | makespan of processing all workloads |
| $f_w$ | total requests of workload $w$ |

(i) A vector $v_c = \{d_n(c)\}_{n=1}^N$ indicating exactly how many GPUs of each type $n$ are used in configuration $c$. (ii) An array $s_c = \{t_1, t_2, \ldots, t_S\}$ indicating the parallelism strategy used in configuration $c$. The array length $S$ represents the total number of pipeline stages, and the element $t_s$ represents the TP degree of the $s$-th stage. The summation of all $t_s$ should be equal to the total GPU count of configuration $c$, i.e., $\sum_{s=1}^{S} t_s = \sum_{n=1}^{N} d_n(c)$. (iii) A cost $o_c = \sum_{n=1}^{N}(d_n(c) \times p_n)$ indicating the total price required for configuration $c$. (iv) A throughput $h_{c,w}$ indicating the rate at which configuration $c$ process workload type $w$, which is obtained through a one-time profiling [1]. By optimizing the configurations, we can obtain the **GPU composition** and **deployment configurations** mentioned in §4.1.

**Workloads and assignment.** Let there be $W$ workload types, indexed by $w \in \{1, 2, \ldots, W\}$. Each workload $w$ must be fully served (i.e., 100% coverage). We allow fractional assignment: a fraction $x_{c,w} \in [0, 1]$ of workload $w$ may be processed by configuration $c$. Concretely, $\sum_{c \in \mathcal{C}} x_{c,w} = 1, \forall w = 1, 2, \ldots, W$. We also introduce an integer variable $y_c \in \{0, 1, 2, \ldots\}$ indicating how many copies of configuration $c$ are chosen (activated). If $y_c = 0$, then $x_{c,w}$ must be zero for all $w$. By co-optimizing the workload assignment (the fractions $x_{c,w}$) with the activated configurations ($y_c$), we determine the final **workload assignment** as described in §4.1.

**Budget and GPU constraints.** A valid configuration set $\mathcal{C}$ must also satisfy the following constraints: (i) the allocated number of GPUs for each type must not exceed the available number, i.e., $0 \leq \sum_{c \in \mathcal{C}}(d_n(c) \times y_c) \leq a_n, \forall n = 1, \ldots, N$; (ii) the total cost of all chosen configurations must be within

---

[1]The detailed procedure of the one-time profiling is demonstrated in Appendix L.

the user-defined budget $B$, i.e., $\sum_{c \in \mathcal{C}} (o_c \times y_c) \leq B$.

**Optimization objective.** We define a makespan variable $T \geq 0$ to represent the overall completion time. For a configuration $c$, if it is instantiated $y_c$ times and processes fractions $x_{c,w}$ of workload $w$, each replica provides a throughput of $h_{c,w}$. Let $f_w$ be the total number of requests for workload $w$. Consequently, the total effective throughput for $c$ is $y_c \times h_{c,w}$, and the time required on $c$ is given by $T_c = \sum_{w=1}^{W} \frac{x_{c,w} \cdot f_w}{y_c \cdot h_{c,w}}$. Since all chosen configurations run in parallel, the system completes once the slowest configuration finishes. Thus, we have $T_c \leq T$ for all $c \in \mathcal{C}$. Our optimization objective is to minimize $T$.

**MILP formulation.** The problem can be summarized as the following Mixed-Integer Linear Program (MILP):

$$\arg\min T \tag{1}$$

$$\text{s.t.} \quad \sum_{c \in \mathcal{C}} x_{c,w} = 1, \, \forall\, w, \textbf{ (Assignment Constraint)} \tag{2}$$

$$\sum_{w \in W} \frac{x_{c,w} \cdot f_w}{y_c \cdot h_{c,w}} \leq T, \, \forall\, c, \textbf{ (Makespan)} \tag{3}$$

$$x_{c,w} \leq y_c, \, \forall\, c, w, \textbf{ (Activation Coupling)} \tag{4}$$

$$\sum_{c \in \mathcal{C}} \big( o_c \times y_c \big) \leq B, \textbf{ (Budget Constraint)} \tag{5}$$

$$\sum_{c \in \mathcal{C}} \big( d_n(c) \times y_c \big) \leq a_n, \, \forall\, n, \textbf{ (GPU Avail.)} \tag{6}$$

$$y_c \in \{0, 1, 2, \dots\}. \tag{7}$$

This formulation determines which configurations are used ($y_c$) and how the workload fractions ($x_{c,w}$) are distributed, subject to memory limit, price budget, and GPU availability constraints, in order to minimize the makespan $T$. Note that $d_n(c)$ is an integer; we enumerate all feasible integer combinations $\{d_n(c)\}_{n=1}^{N}$ in a precomputation step. In contrast, $x_{c,w}$ is a continuous variable, and the solver relies on branch-and-bound to systematically narrow the feasible region and converge to an optimal fractional assignment.

**Complexity analysis.** The number of binary activation variables $y_c$ grows combinatorially with the number of feasible configurations $|\mathcal{C}|$. In this worst case $|\mathcal{C}|$ can be on the order of $\prod_{n=1}^{N}(a_n + 1)$. Since MILP solvers (e.g., branch-and-bound) have worst-case running time exponential in the number of binary variables, the theoretical worst-case time complexity scales as $\mathbb{O}\big(\text{poly}(|\mathcal{C}|, W, N) \times 2^{|\mathcal{C}|}\big)$, where the polynomial factor accounts for the overhead of processing each node in the search tree (e.g., solving continuous relaxations of the MILP). As a result, the solution time escalates rapidly with the number of candidate configurations.

**Other constraints and heuristics.** We introduce additional constraints and heuristics to reduce the search space. Concretely, we enforce a memory constraint to eliminate configurations with insufficient GPU memory, and a connectivity constraint to exclude those involving disconnected GPUs. Additionally, we refine the configuration search by restricting TP to single machines and enabling non-uniform PP

layer partitioning based on memory allocation. See Appendix D for details.

**Binary search.** To address the long computation times associated with the MILP solver for large-scale problems, we introduce a **binary-search-on-T** method to accelerate the search process. Rather than directly minimizing the makespan $T$, we iteratively check whether a valid serving plan exists for different candidate values of $\hat{T}$, based on reasonable lower and upper bounds. For a full explanation, refer to Appendix F, and we evaluate the effectiveness of this binary search method in §5.2.

**Extension to multiple LLM serving.** Our MILP formulation can be easily adapted to scenarios involving multiple LLMs, such as simultaneously serving both Llama3-8B and Llama3-70B models. To accommodate this, we introduce a model-type dimension to the decision variables and constraints. This ensures that workload assignments, memory requirements, and other constraints are optimized for each model type. The objective remains to minimize the overall makespan $T$, while also taking into account GPU availability, budget constraints, and other constraints across all model types. For a detailed explanation of the formulation, please refer to Appendix E. We demonstrate the evaluation of our method for multiple model serving in §5.2.

## 5. Experiments

### 5.1. Experimental Setup

**Environments.** Our experiments are conducted using two types of data center servers H100 and A100, three types of work station servers A40, RTX A6000 and L40, and one type of consumer server RTX 4090. In data center servers, GPUs are linked by NVLink (300 GB/s), while in workstation/consumer servers, GPUs are linked by PCIe (60 GB/s). Servers with inter-connection are connected via Ethernet with a bandwidth of 5 Gb/s. All experiments are conducted with vLLM (Kwon et al., 2023).

**Baselines.** We compare our method, which uses heterogeneous cloud resources, against various homogeneous setups:

- **Heterogeneous setups:** We rent GPUs from Vast.ai, a cloud provider offering a range of GPU types. The rentals are based on real-time GPU availability on the cloud. For our experiments, we randomly selected four different GPU availabilities (shown in Table 4 in Appendix H) under varying price budgets of 15, 30, and 60 \$/h.

- **Homogeneous setups:** We rent H100 GPUs (representative data center GPUs), RTX A6000 GPUs (workstation GPUs), and RTX 4090 GPUs (consumer GPUs) under different price budgets, with each GPU type representing a homogeneous baseline. For example, a budget of 60 \$/h can rent up to 20 H100 GPUs. Note that we fine-tune the deployment configurations and workload assignments us-

ing our scheduling algorithm to optimize the performance of each homogeneous baseline.

**Models and traces.** Our evaluation is conducted on Llama3-8B and Llama3-70B models. And we follow prior work to generate workload traces based on real-world data. Our testing traces are subsampled from three sources: real workload traces collected over one month from the Swiss AI Center, the WildChat dataset, and the production traces Azure-Trace. Each trace comprises multiple workload types, with their ratios shown in Table 5 in Appendix I.

**Evaluation metrics.** We focus on the overall system throughput and vairous percentile latencies (i.e., p10, . . . , p90, p100). P90 latency represents the maximum response time within which 90% of all requests are completed.

## 5.2. Experimental Results

**End-to-end system performance.** We evaluated our method across various traces, cloud GPU availability scenarios, price budgets, model types, and homogeneous baselines. Experimental results in Figure 5 and Figure 6 show that our method improves system throughput by up to 41% (25% on average) while reducing percentile latencies by up to 54% (20% on average). In traces 1 and 2, H100 (Homo) achieves the best performance among all baselines. In our plan, the GPU composition depends on the budget. With a high budget (60 \$/h), data center GPUs are preferred, making up approximately 51% of our rented resources for request processing. In contrast, with a low budget (15 \$/h), workstation GPUs are favored due to their lower cost. In trace 3, A6000 (Homo) demonstrates the highest performance among all baselines. In this scenario, our plan primarily relies on workstation GPUs, which constitute approximately 93% of the rented resources for request processing. Additionally, as shown in Figure 15 in Appendix J, the 4090 (Homo) delivers the best performance among all baselines for the Llama3-8B model. In this case, our plan prefers consumer GPUs, which form the majority of our rented resources and handle 53% of overall request processing.

**Comparison with HexGen.** We also compare our method with the state-of-the-art heterogeneous serving framework, HexGen (JIANG et al.). Since HexGen schedules workloads based on a fixed GPU composition, we evaluate it using two setups: (i) a uniform composition, where six GPU types are evenly allocated within the budget, and (ii) the optimal composition used by our method. As shown in Figure 7, HexGen with a uniform composition suffers up to 35% and on average 29% performance degradation due to suboptimal GPU allocation. Even with the optimal composition, our method outperforms HexGen by up to 18% and on average 14%, benefiting from workload-aware scheduling.

**Compare with Helix.** We conduct additional experiments

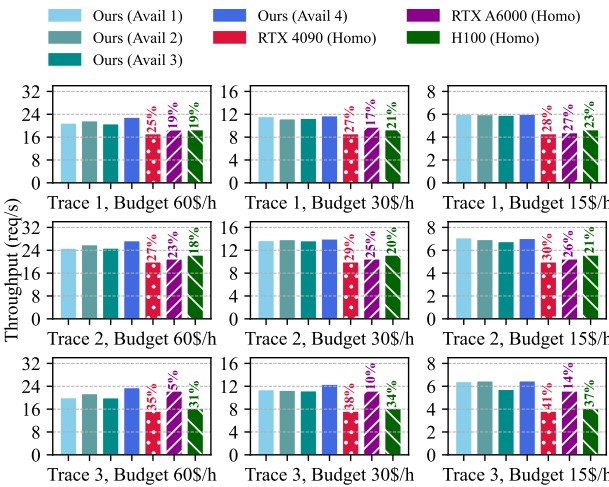

Figure 5: End-to-end throughput results on Llama3-70B model with different setups. We further demonstrate the Llama3-8B results in Appendix J.

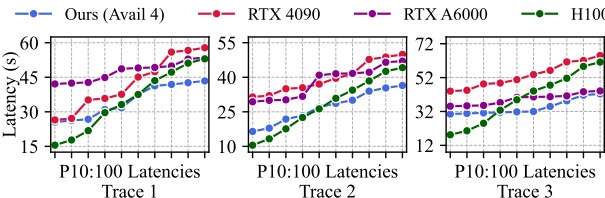

Figure 6: End-to-end latency results on Llama3-70B model with different setups.

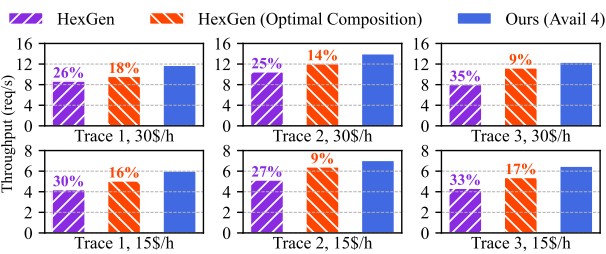

Figure 7: Ours vs. HexGen. The first and second bars in each picture represent HexGen using a uniform and optimal GPU composition.

comparing our system with Helix (Mei et al., 2024). Specifically, we compare our method against Helix's single-cluster case (the optimal case reported in their paper) under a price budget of \$15 per hour on the AzureTrace dataset (i.e., trace 2). While Helix optimizes heterogeneous LLM serving using max-flow and MILP scheduling, our method explicitly considers workload heterogeneity and GPU composition optimization, resulting in greater cost efficiency. Experimental

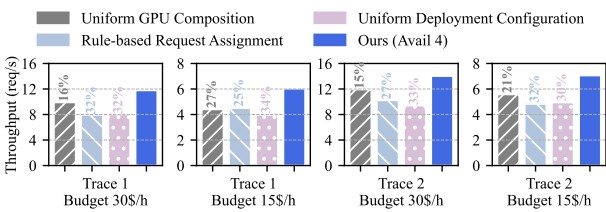

Figure 8: Ablation study of Llama3-70B on traces 1 and 2.

results show that our system outperforms Helix by 25–35%.

Table 3: Performance comparison of Helix and our method.

|       | **Llama-30B**      | **Llama3-70B**    |
|-------|--------------------|-------------------|
| Helix | 8.49 req/s         | 5.72 req/s        |
| Ours  | 11.43 req/s (35%↑) | 7.13 req/s (25%↑) |

**Ablation studies.** We assess the impact of each optimization target in our scheduling algorithm by systematically disabling them. Three baselines are considered: (i) Uniform GPU composition: GPUs are rented uniformly across six types within the given budget. This evaluates the performance gains from optimized heterogeneous GPU composition. (ii) Uniform deployment configuration: Instead of optimizing deployment per model replica, TP is uniformly applied across all replicas. This measures the impact of deployment configuration optimization. (iii) Rule-based request assignment: Requests are assigned using a Round-Robin strategy based on real trace arrivals, assessing the benefit of heterogeneous-aware workload assignment. As shown in Figure 8, disabling heterogeneous GPU composition reduces throughput by up to 27% (average: 20%), deployment optimization by up to 34% (average: 33%), and workload assignment by up to 32% (average: 29%). These results highlight the necessity of each optimization for high-performance LLM serving in heterogeneous environments.

**Algorithm efficiency.** We evaluate two strategy search methods from §4: (i) MILP and (ii) binary search. As shown in Figure 9, the left plot illustrates their scalability, while the right plot depicts algorithm performance during the search process. Compared to MILP, which exhaustively explores all combinations of heterogeneous GPU compositions, deployment configurations, and workload assignments, the binary search method, enhanced with feasibility checks using knapsack approximation, achieves approximately a 4× reduction in search time. This improvement comes with only marginal differences in algorithm performance, with deviations of less than 1%.

**Multi-model extension.** We further evaluate our system in a multi-model serving scenario (discussed in §4.3), assuming that 80% of the requests are assigned to the Llama3-

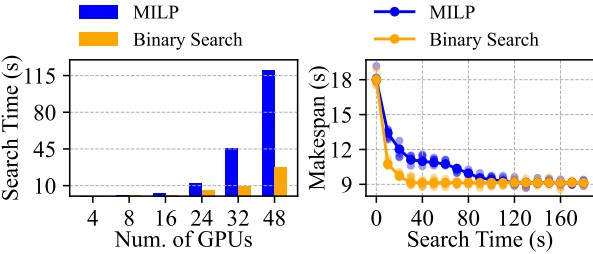

Figure 9: Algorithm scalability and efficiency.

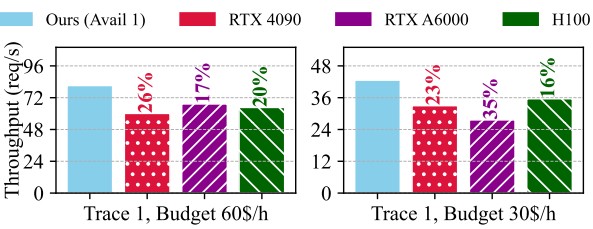

Figure 10: End-to-end experiments on multiple model types (Llama3-8B and Llama3-70B) with different setups.

8B model, while the remaining 20% are assigned to the Llama3-70B model. As shown in Figure 10, our method outperforms homogeneous baselines, achieving up to a 35% (average: 23%) performance gain. In the 60 $/h case, our scheduling algorithm allocates 70% of computing resources to Llama3-70B and 30% to Llama3-8B. In the 30 $/h case, the allocation shifts to 77% and 23%. Our algorithm balances resource allocation based on model demands, enabling efficient multi-model serving in heterogeneous clusters.

**System performance vs. price budget.** We compare our system with homogeneous baselines under different price budgets. The performance gap narrows as the budget increases due to cloud resource limits, which is reasonable since we assume an unlimited number of GPUs for our homogeneous baselines. See Appendix K for details.

**Additional discussion and experiments.** We provide further discussion and experiments on online scheduling for dynamic workloads, as well as on the trade-offs between cost-efficiency and request latency, in Appendix M.

## 6. Conclusion

This paper aims to address the questions of *why* and *how* heterogeneous cloud resources can be utilized for cost-efficient LLM serving. Specifically, we benchmark the cost-efficiency of LLM serving over heterogeneous GPUs, following which, a novel scheduling algorithm is developed. Experimental results demonstrate that our approach outperforms existing works substantially.

## Impact Statement

This paper presents work whose goal is to advance the field of Machine Learning. There are many potential societal consequences of our work, none which we feel must be specifically highlighted here.

## Acknowledgment

This work is supported by the HKUST startup grant R9895 from CSE; RGC-ECS project 26218024; RGC-NSFC project CRS_HKUST601/24. This work was supported as part of the Swiss AI Initiative by a grant from the Swiss National Supercomputing Centre (CSCS) on Alps.

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

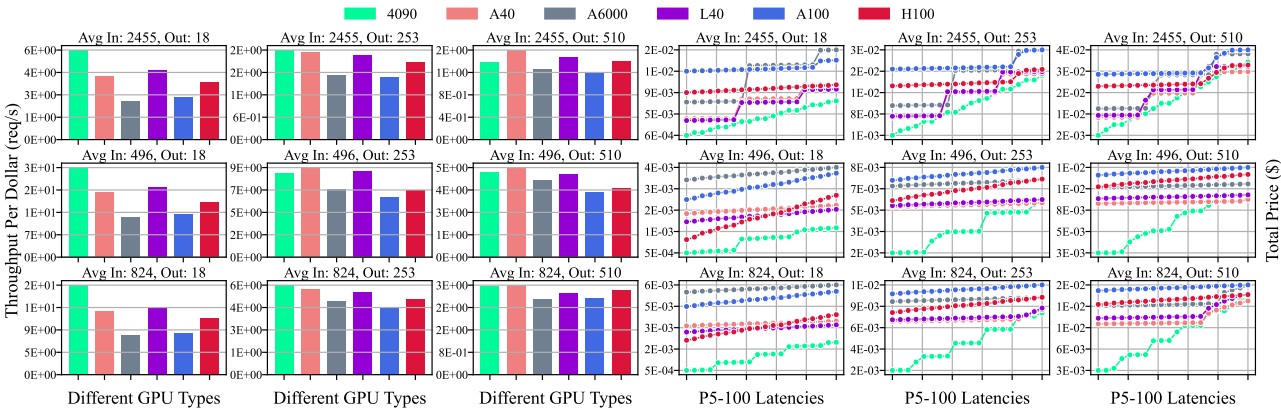

Figure 11: Benchmarked results for Llama3-8B model with different GPU types on different workload types.

## A. Benchmarking Results for Llama3-8B

We demonstrate the benchmark results for Llama3-8B model in Figure 11.

## B. Benchmarking Results of Different Deployment Configurations for Remaining GPUs

We demonstrate the benchmark results for different deployment configurations in Figure 12 and Figure 13.

## C. Simple Example

**Experiment setup.** We begin by assuming three GPU types, $\{t_1, t_2, t_3\}$, each with two units available. The hourly rental prices for these types are 4, 2, and 2 \$/h, respectively. We consider two workload types $\{w_1, w_2\}$, which arrive simultaneously with 80 total requests for $w_1$ ($\lambda_1 = 80$) and 20 total requests for $w_2$ ($\lambda_2 = 20$). We denote by $C_{t,w}$ the throughput (in requests per second) of GPU type $t$ on workload $w$. If each GPU serves one model replica, the throughputs are $C_{1,1} = 1.0$, $C_{1,2} = 1.2$, $C_{2,1} = 0.9$, $C_{2,2} = 0.9$, $C_{3,1} = 0.3$, and $C_{3,2} = 0.5$. Note that $C_{\sim,1}$ and $C_{\sim,2}$ vary with model parallelism. In **Cases 1** and **2**, we assume the workload is assigned to each GPU in proportion to its processing rate, so the system-wide throughput for each workload is the sum of individual-GPU rates. In **Case 3**, we allow workload-aware assignment for further optimization.

**Case 1: GPU composition.** We compare two compositions under the same budget of 8 \$/h, where each GPU is responsible for serving one model replica. Composition 1 uses $1 \times t_1$, $1 \times t_2$, and $1 \times t_3$. This setup achieves a total throughput of $(1.0 + 0.9 + 0.3) = 2.2$ rps on $w_1$ and $(1.2 + 0.9 + 0.5) = 2.6$ rps on $w_2$, giving a processing time of $(\lambda_1/C_{\sim,1} + \lambda_2/C_{\sim,2}) = (80/2.2 + 20/2.6) \approx 44.05$ s. Composition 2 uses $1 \times t_1$ and $2 \times t_2$, for throughputs of $(1.0 + 0.9 + 0.9) = 2.8$ rps on $w_1$ and $(1.2 + 0.9 + 0.9) = 3.0$ rps on $w_2$, so $(80/2.8 + 20/3.0) \approx 35.24$ s. In this case, changing the GPU composition under the same price budget results in a 20% speedup.

**Case 2: Deployment configuration.** Focusing on composition 2, we compare two ways to organize these three GPUs. Configuration 1 keeps all GPUs in a purely DP style (i.e., each GPU is responsible for serving one model replica), summing up to 2.8 rps on $w_1$ and 3.0 rps on $w_2$, matching the 35.24 s above. Configuration 2 applies TP to the two $t_2$ GPUs, which changes their combined rate, e.g., to 2.4 rps on $w_1$ and 1.5 rps on $w_2$. Together with the single $t_1$ GPU (1.0 rps on $w_1$ and 1.2 rps on $w_2$), the total throughput becomes $(3.4, 2.7)$ rps for $(w_1, w_2)$. The corresponding time $(80/3.4 + 20/2.7) \approx 30.94$ s. In this case, changing the deployment configuration results in an improvement in the overall processing time of roughly 14%.

**Case 3: Workload assignment.** Finally, we keep the same composition and TP-based configuration but allow workload-aware assignment. Concretely, we assign:

$$\text{Replica } (t_1): 15\% \text{ of } w_1, 100\% \text{ of } w_2,$$
$$\text{Replica (TP on } 2 \times t_2): 85\% \text{ of } w_1.$$

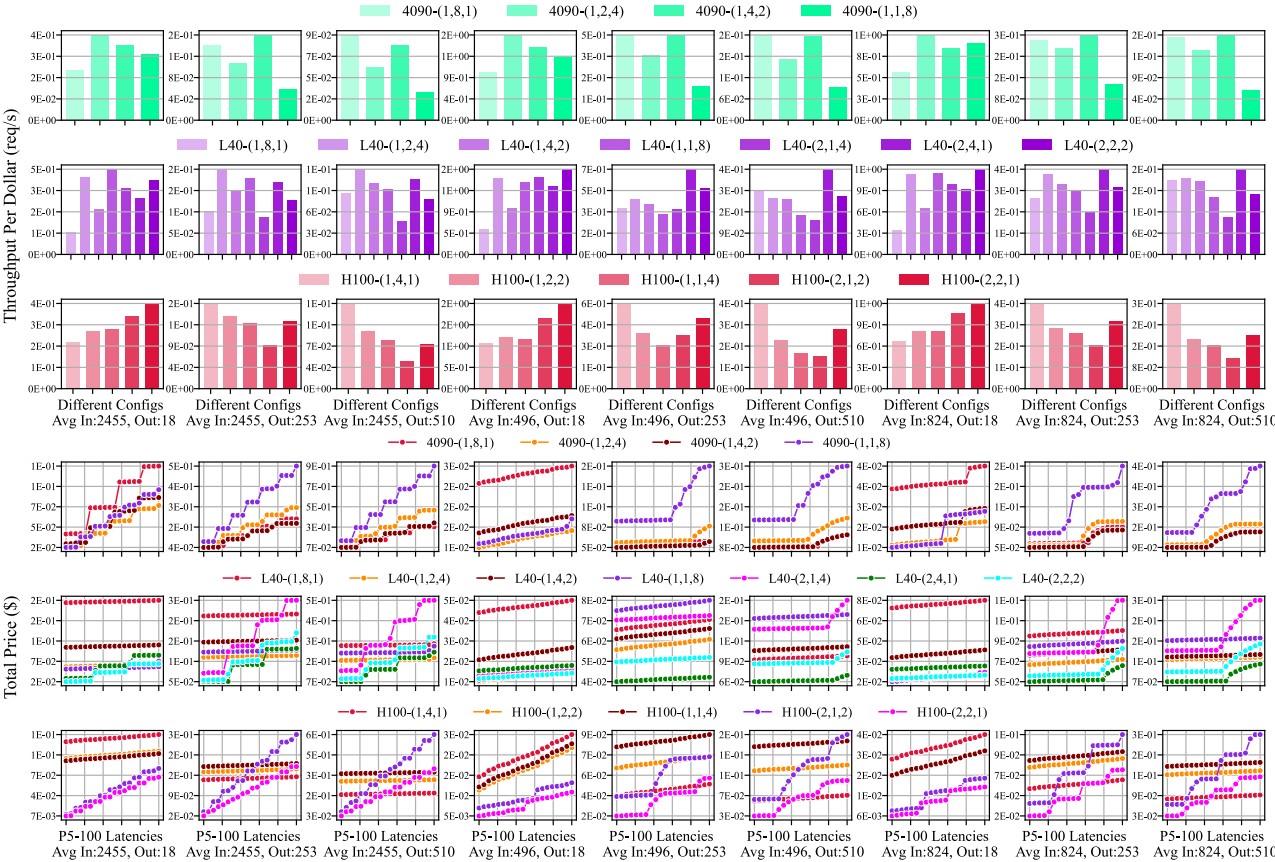

Figure 12: Throughput and latency results for Llama3-70B model with different deployment configurations on different workloads.

Under these fractions, $t_1$ processes 12 requests of $w_1$ at 1.0 rps and 20 requests of $w_2$ at 1.2 rps, while the TP-based replica handles 68 requests of $w_1$ at 2.4 rps. By balancing the load and routing the workload to the preferable replica, i.e., the one with relatively higher throughput for a specific workload, we reduce the overall completion time from 30.94 s to $max(0.85\lambda_1/C_{2,1}, 0.15\lambda_1/C_{1,1} + \lambda_2/C_{1,1}) = max(68/2.4, 12/1 + 20/1.2) = 28.67$ s. In this case, changing the workload assignment results in an additional improvement in the overall processing time of approximately 8%.

This step-by-step example (also illustrated in Figure 14) shows how all three factors—GPU composition, deployment configuration, and workload assignment—must be jointly optimized to achieve the best performance.

## D. Other Constraints and Heuristics

We enforce two additional constraints to minimize the overall search space and speed up the search process: (i) We perform an early memory check on each configuration, which ensures that the sum of GPU memories in configuration $c$ is sufficient for a model replica, i.e., $\sum_{n=1}^{N}(d_n(c) \times m_n) \geq M_r$, where $M_r$ represents the least memory required for serving one model replica (e.g., 140 GB for Llama3-70B model). Configurations that violate this constraint will be eliminated from further evaluation; (ii) we enforce a connectivity constraint within each configuration. If certain GPUs lack interconnection (e.g., they are located in different data centers), those combinations do not appear in each configuration $c$. Additionally, we use two heuristic methods to facilitate the deployment configuration search: (i) we only adopt TP within a single machine containing multiple GPUs, as TP typically requires high intra-machine communication bandwidth (e.g., PCIe, NVLink) for efficient deployment; (ii) we support non-uniform pipeline layer partitioning for PP, and determine the partition based on the total memory allocated for each stage. For instance, if there are a total of 24 layers and the GPU memory allocated for each stage is 1:2, then we allocate 8 and 16 layers to the first and second stages.

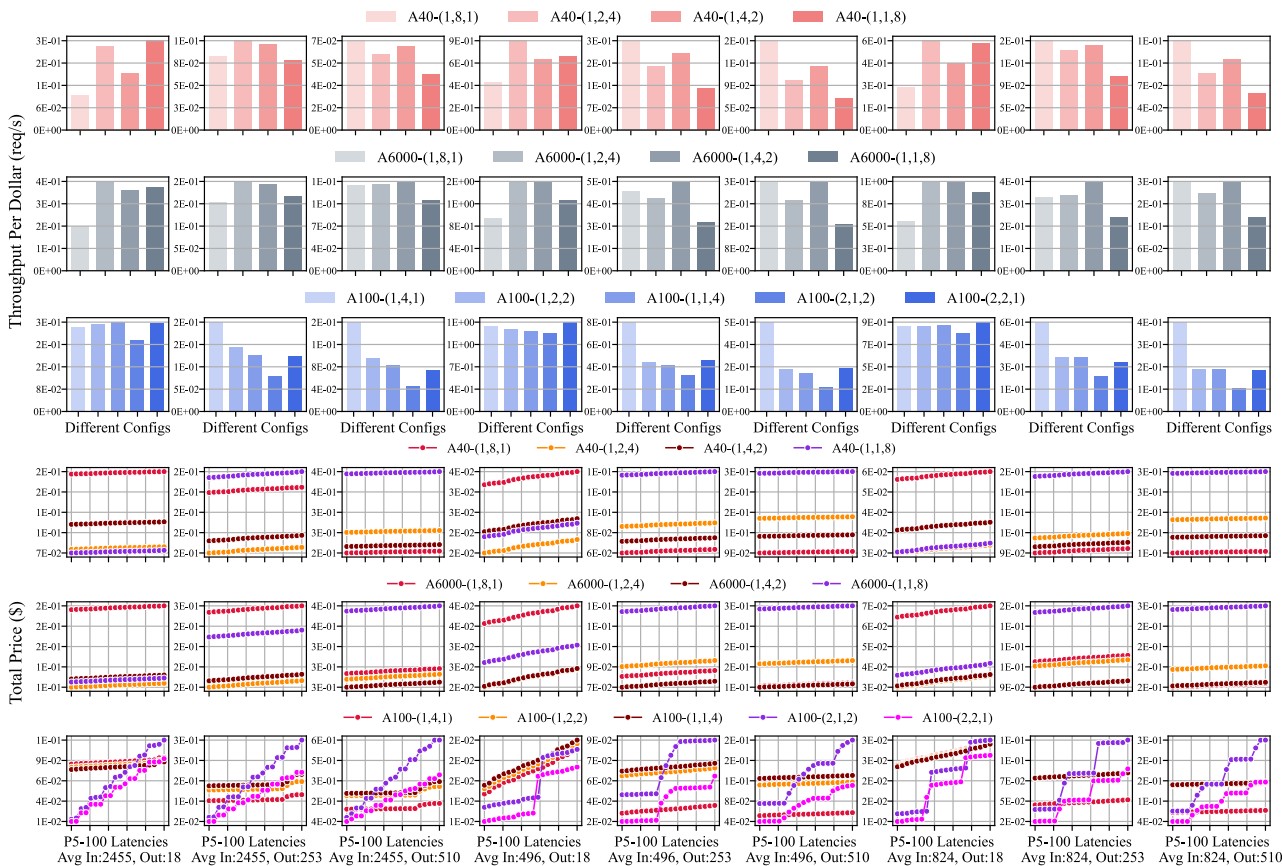

Figure 13: Throughput and latency results for Llama3-70B model with different deployment configurations on different workloads.

## E. Extend to Multiple LLM serving

The previous MILP formulation assumes a single LLM serving with multiple model replicas. However, cloud services typically involve multiple LLM serving with varying sizes, e.g., Llama3-8B and Llama3-70B models. To integrate multiple LLM serving plan search into our MILP, we introduce the following extended MILP formulation.

Let there be $M$ model types, indexed by $m \in \{1, 2, \ldots, M\}$, each type has its own memory requirement. The MILP formulation can be extended to:

$$\arg\min T \tag{8}$$

$$\text{s.t.} \quad \forall m : \begin{cases} \sum_{c \in \mathcal{C}_m} x_{c,w,m} = 1, \, \forall \, w \in W_m, \\ \sum_{w \in W_m} \frac{x_{c,w,m} \cdot f_{w,m}}{y_{c,m} \cdot h_{c,w,m}} \leq T, \, \forall \, c \in \mathcal{C}_m, \\ x_{c,w,m} \leq y_{c,m}, \, \forall \, c \in \mathcal{C}_m, \, \forall \, w \in W_m, \end{cases} \tag{9}$$

$$\sum_{m=1}^{M} \sum_{c \in \mathcal{C}_m} \left( o_{c,m} \times y_{c,m} \right) \leq B, \tag{10}$$

$$\sum_{m=1}^{M} \sum_{c \in \mathcal{C}_m} \left( d_n(c,m) \times y_{c,m} \right) \leq a_n, \, \forall \, n, \tag{11}$$

$$y_{c,m} \in \{0, 1, 2, \ldots\}. \tag{12}$$

In this extended MILP formulation, we introduce an additional model-type dimension to every relevant variable and constraint. Consequently, the problem now accommodates multiple model types (each with its own workload set, throughput profiles, memory requirements, etc.) within a unified optimization framework. The objective remains the same—minimizing

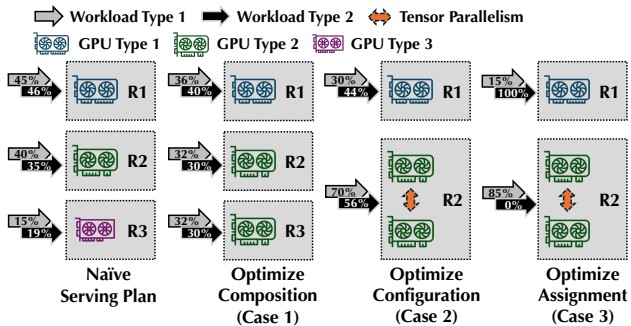

Figure 14: Illustration of a simple example.

---

**Algorithm 1** Binary Search on $T$

---

**Input:** $\underline{T}, \overline{T}$ {initial bounds}
**Input:** $\tau$ {tolerance}
**Output:** Approximate minimal feasible makespan
**while** $(\overline{T} - \underline{T}) > \tau$ **do**
$\quad \hat{T} \leftarrow \frac{\underline{T}+\overline{T}}{2}$
$\quad$**if** FEASIBILITYCHECK($\hat{T}$) **is true then**
$\quad\quad \overline{T} \leftarrow \hat{T}$ {If feasible, try smaller $\hat{T}$}
$\quad$**else**
$\quad\quad \underline{T} \leftarrow \hat{T}$ {If infeasible, increase $\hat{T}$}
$\quad$**end if**
**end while**
**return** $\overline{T}$

---

the overall makespan $T$—while jointly enforcing GPU availability, budget, and other constraints across all model types. This ensures that the chosen configuration set and workload assignments meet the demands of every model type while adhering to the total GPU and budget limits.

## F. Binary Search

For large numbers of model, workload and GPU types, it might take hours for the MILP solver to provide a relatively good solution. To expedite the search process, we incorporate the **binary-search-on-T** approach into our existing MILP formulation. Specifically, we transform the previous "minimize $T$" problem into a sequence of feasibility checks: for a given candidate $\hat{T}$, we ask whether a valid serving plan exists that completes all workloads in at most $\hat{T}$, subject to budget and GPU constraints. If yes, we can try smaller $\hat{T}$; if no, we must increase $\hat{T}$.

**Binary search.** The lower bound of the makespan, $\underline{T}$, is identified as the best possible time if infinite GPUs were available with no budget limit (e.g., using the fastest configuration for each workload type). The upper bound, $\overline{T}$, is the worst-case scenario (e.g., using the slowest feasible configuration to serve all workloads). During the binary search loop, if the difference between the lower and upper bounds exceeds a certain tolerance $\tau$ (e.g., one second), i.e., $\overline{T} - \underline{T} \geq \tau$, we calculate $\hat{T} = \frac{\overline{T}+\underline{T}}{2}$ and check its feasibility. If a pair $(x_{c,w}, y_c)$ or $(x_{c,w,m}, y_{c,m})$ (in the extended case) satisfies all constraints in §4.3 or §4.3, with $\sum_{w \in W} \frac{x_{c,w}}{h_{c,w}} \leq \hat{T}$ or $\sum_{w \in W_m} \frac{x_{c,w,m}}{h_{c,w,m}} \leq \hat{T}, \forall c \in C_m$, we update $\overline{T} \leftarrow \hat{T}$. Otherwise, we update $\underline{T} \leftarrow \hat{T}$. When the loop concludes, the value of $\overline{T}$ (or $\underline{T}$) represents the minimal feasible makespan within the specified tolerance. Note that the feasibility check can be further approximated using a knapsack approximation, which makes the binary search approach more efficient for handling large-scale MILP problems. We outline the binary search process in Algorithm 1.

**Other optimizations for speeding up MILP.** For extremely large-scale MILP problems (e.g., dozens of model and workload types with hundreds of GPUs), we introduce several optimizations, such as pruning configurations, providing a good starting point, and early stopping based on the lower bound, as detailed in Appendix G. The experimental results presented in §5

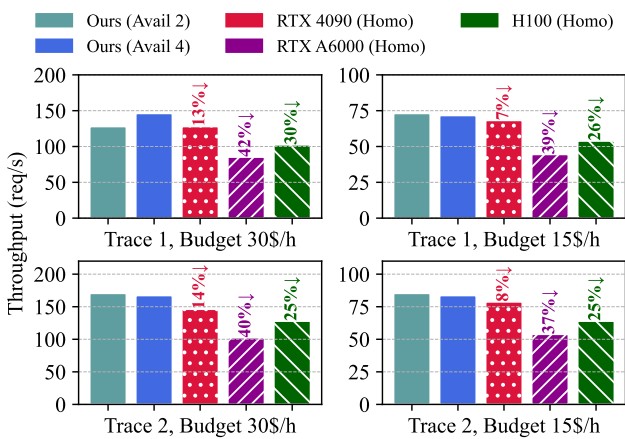

Figure 15: End-to-end experiments on Llama3-8B model with different setups.

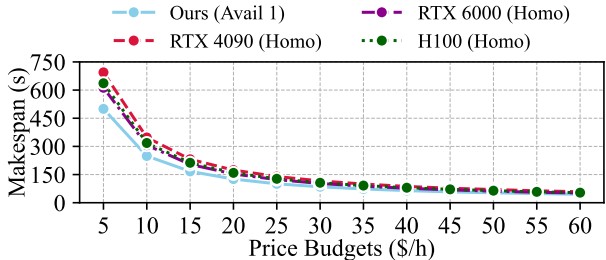

Figure 16: System performance v.s. price budget.

demonstrate the efficiency, effectiveness, and scalability of our scheduling algorithm.

## G. Other Optimizations for Speeding up MILP

For large numbers of GPUs and model types, it might take hours for the MILP solver to provide a relatively good solution. To expedite the search process, we introduce three optimizations to minimize the search space without sacrificing the effectiveness of our scheduling results: (i) for each model type, we prune configurations that are clearly dominated. For example, configurations with high degrees of model parallelism are retained for Llama3-70B, which requires substantial memory for model serving, but are pruned for Llama3-8B to prevent excessive communication overhead; (ii) we pre-estimate the resource requirements for each model type based on incoming workloads and their memory demands, and proportionally allocate resources to provide a good starting point for the MILP solver, thereby expediting the search process; (iii) we establish a theoretical lower bound for the makespan by analyzing the minimum possible processing time across all feasible configurations, which enables the implementation of early stopping criteria during optimization, i.e., the search process stops when it finds a solution that is very close to this lower bound. The minimum possible makespan occurs when all workloads are assigned to the most efficient configuration without considering resource constraints.

## H. Real Time GPU Availabilities

We randomly selected four real-time GPU availabilities on the cloud, as shown in Table 4.

## I. Workload Type Ratios for Each Trace

We demonstrate the workload type ratios for the three traces in Table 5.

Table 4: Real time GPU availabilities on cloud platform.

| GPU Avails | 4090 | A40 | A6000 | L40 | A100 | H100 |
|---|---|---|---|---|---|---|
| Avail 1 | 16 | 12 | 8 | 12 | 6 | 8 |
| Avail 2 | 32 | 8 | 16 | 16 | 7 | 12 |
| Avail 3 | 32 | 16 | 8 | 8 | 32 | 8 |
| Avail 4 | 24 | 24 | 24 | 16 | 4 | 8 |

Table 5: Workload type ratios for subsampled traces from the Swiss AI Center (Trace 1), Azure-Trace (Trace 2), and WildGPT dataset (Trace 3). Workloads 1–9 correspond to the nine workload types shown in Figure 4 from left to right.

| Workloads | 1 | 2 | 3 | 4 | 5 | 6 | 7 | 8 | 9 |
|---|---|---|---|---|---|---|---|---|---|
| Trace 1 (%) | 33 | 7 | 8 | 7 | 27 | 6 | 6 | 3 | 3 |
| Trace 2 (%) | 22 | 5 | 5 | 21 | 5 | 5 | 19 | 6 | 12 |
| Trace 3 (%) | 4 | 1 | 4 | 3 | 20 | 27 | 1 | 25 | 15 |

## J. End-to-end Experiment Results for Llama3-8B Model

The end-to-end experiments on Llama3-8B model with different setups are shown in Figure 15.

## K. System Performance vs Price Budget

We further evaluate our system's performance compared to homogeneous baselines under various price budgets. As shown in Figure 16, as the price budgets increase (from 5 \$/h to 60 \$/h), the performance gap between our approach and the homogeneous setups narrows from approximately 30% to 15%. This is primarily due to the limited availability of cloud resources. In homogeneous baselines, we assume an unlimited number of GPUs, allowing performance to scale linearly with the price budget. However, in cloud-based scenarios, resource restrictions prevent such linear scaling. When larger price budgets are applied, unsuitable GPUs for the current workload may be rented if they are the only available options, further limiting performance scalability.

Table 6: Performance of different configurations.

| Diff Configs | Real | Estimated |
|---|---|---|
| H100 (2,4) | 0.56 req/s | 0.60 req/s |
| H100 (4,2) | 0.44 req/s | 0.47 req/s |
| H100 (4,2) (cross machine) | 0.42 req/s | 0.44 req/s |
| L40 (2,4) | 0.42 req/s | 0.46 req/s |
| L40 (4,2) | 0.21 req/s | 0.22 req/s |
| L40 (4,2) (cross machine) | 0.18 req/s | 0.19 req/s |
| H100+A100 (4,2) (cross machine) | 0.48 req/s | 0.52 req/s |

## L. One-Time Profiling

We employ a one-time profiling strategy that captures the following components. This approach is referred to the profiling method used in Vidur (Agrawal et al., 2024a): (1) Inference prefilling latency: We profile the latency for a single transformer layer across varying TP degrees, different workload types, and various GPU types; (2) inference decoding latency: We profile the decoding latency for a single transformer layer under similar variations in TP degrees, workload types, and GPU types; (3) pipeline communication latency: We measure the communication latency between different GPUs across various workload types. Using these measurements, the per-request latency for any configuration is estimated by combining the TP costs (both communication and computation) of all layers—which may be served by different GPUs and at varying TP degrees—with the PP communication cost. Note that our heuristics, as discussed in Section 4.3 and Appendix D, largely reduce the profiling space, e.g., TP is employed only intra-machine. When estimating throughput, the prefill and decoding

phases are treated separately: (1) The prefill phase is compute-bound, and its batched processing capacity is determined by the sum of the individual latencies; (2) the decoding phase is memory-bound, with its batched processing capability defined by a single latency value. This distinction has been validated in several studies (Zhong et al., 2024; Patel et al., 2024).

Table 6 demonstrates examples of our cost estimation under a long-input, short-output workload (i.e., workload 1 in Figure 4). In Table 6, the notation (2,4) indicates that the TP degree is 2 and the PP degree is 4. The estimation errors range from 4% to 7%. Although the estimations are not perfectly accurate, they are sufficiently reliable for selecting the optimal configurations.

## M. Discussion

**Online replanning.** Online scheduling for dynamic workloads is orthogonal to the approach presented in this work. However, accommodating dynamic workloads could be achieved through the implementation of a replanning mechanism analogous to the one proposed in DistServe (Zhong et al., 2024). Concretely, the system could (1) monitor the real-time composition of incoming workloads, (2) track GPU resource availability within the cloud environment, and (3) upon detecting a significant shift in workload distribution, (e.g., an increase in the proportion of certain workload types) the scheduling algorithm could be executed again, incorporating recent historical data to produce an updated serving plan.

Table 7: Performance changes in workload and GPU drop.

|  | **Workload Change** | **GPU Drop** |
|---|---|---|
| Before | 26.89 req/s | 26.89 req/s |
| After | 23.70 req/s (13%↓) | 20.80 req/s (29%↓) |
| Replanning | 29.61 req/s (25%↑) | 22.85 req/s (10%↑) |

As shown in Table 7, we test the workload surge in short output requests in AzureTrace dataset (i.e., trace 2) with a price budget of $30 per hour. Before surge, the optimal GPU composition is {20%, 65%, 15%} for datacenter, workstation, and consumer GPUs, achieving 26.89 req/s. After workload change, the throughput degrades to 23.7 req/s. In this case, replanning (shifting allocation to {63%, 23%, 14%}) boosts throughput to 29.61 req/s. We also test the case when GPU drop happens (4 H100s down), the throughput falls from 26.89 to 20.80 req/s. In this case, replanning raises throughput to 22.85 req/s.

**Trade-offs between prioritizing cost-efficiency and request latency.** Prioritizing cost-efficiency typically involves using fewer resources (i.e., lower budgets), which can lead to slightly higher response latencies. In contrast, prioritizing latency often requires utilizing more resources (i.e., incurring higher costs). We acknowledge that optimizing for cost efficiency may result in a slight increase in latency. However, inference tasks typically do not require extremely low latency; meeting a predefined latency threshold is usually sufficient. In resource-limited scenarios, where systems are naturally under-provisioned, emphasizing throughput can also indirectly improve latency by reducing queuing delays. Our experimental results in Figure 6 demonstrate that our method achieved the lowest P99 latency among all compared baselines.

