# OpenReview forum: "Demystifying Cost-Efficiency in LLM Serving over Heterogeneous GPUs"
_ICML.cc/2025/Conference — ICML 2025 poster_

### Official Review · Reviewer_2VxX · 2025-03-12

**Overall Recommendation:** 3

**Summary:**

The paper investigated a highly practical problem of co-optimizing GPU composition, deployment configurations, and workload assignment in a heterogeneous GPU environment under a budget constraint to maximize Large Language Model (LLM) serving efficiency. The authors proposed a Mixed Integer Linear Programming (MILP) formulation to optimize the makespan for processing all workloads. Experimental results showed that the proposed method outperforms both homogeneous and heterogeneous baselines across a wide range of scenarios.

**Claims And Evidence:**

- The paper claims that LLM serving efficiency can be significantly influenced by heterogeneous GPU composition, deployment configuration, and workload assignment policy. This claim is well supported by the evidence in Figures 3 and 4, as well as the example provided in Section 4.2.

- The paper claims that the proposed method significantly outperforms both homogeneous and heterogeneous baselines across a wide range of scenarios. This claim is generally supported by the experimental results.

**Essential References Not Discussed:**

No

**Experimental Designs Or Analyses:**

The experimental design is generally well-structured, and the results are thoroughly analyzed. The authors compare their system's end-to-end performance against the primary baseline, HexGen, and conduct ablation studies to assess the effects of GPU composition, deployment configurations, and request assignment policies. In addition, they evaluate the effectiveness of their proposed MILP problem heuristic. Methods for extending to multi-model cases are also discussed.

However, HexGen is currenty the only baseline used for comparison. Introducing more baselines would provide a stronger support for the paper's performance claim.

**Methods And Evaluation Criteria:**

The method proposed in this paper has several significant problems.

- In the formulated MILP problem, a deployment configuration $c$ is defined by a tuple $(v_c, s_c, o_c, h_{c,w})$, with $h_{c,w}$ defined as the "throughput of config $c$ on workload $w$", and $v_c$ indicating the GPU composition. However, I find this questionable, as the throughput should be affected by the GPU composition. This would mean $h_{c,w}$ is recursively defined. Here, it seems that the authors are treating the throughput as invariant to the GPU composition, which is not realistic.

- Also for the $h_{c,w}$, solving the MILP problem requires it to be known in advance. This is a significant limitation, as measuring this throughput requires non-trivial effort and resources, especially considering the vast number of possible configurations and workloads.

- The MILP formulation aims to optimize the makespan for processing all workloads. However, the makespan is affected by the distribution of workloads, a factor entirely absent from the MILP formulation.

**Other Comments Or Suggestions:**

I have some very specific comments and suggestions for the authors to consider, listed below:

- The y-axis label "Throughput" for the left subplots in Figure 3 may mislead readers, as it doesn’t reflect cost efficiency. I would suggest "Throughput Per Dollar" instead.
- Also in Figure 3, the x-axis label "P5-P100 Latencies" for the right subplots needs a better explanation. Exactly what percentiles are being referred to here?
- The Figure 4 is difficult to interpret.
  - I recommend explicitly stating what the bars in each subplot represent. The current x-axis labels seem incorrect and should likely be "Different Deployment Configurations" instead.
  - Additionally, Observation 2 from the paper is not apparent in this figure.
- In the "Workloads and Assignment" paragraph of Section 4.3, the symbols $m$ and $M$ appear to be errors and should be corrected to $w$ and $W$, respectively.
- The reference format requires substantial revision. Several cited works, published in conference proceedings, are inaccurately listed as arXiv preprints, which is misleading and needs correction.

**Other Strengths And Weaknesses:**

- The paper's overall presentation is generally clear and well-organized.
- This paper focuses on a highly practical problem, exploring the challenges of optimizing LLM serving efficiency in a heterogeneous GPU environment under budget constraints.
- Despite the questionable workload distribution setup, the experimental results are generally well-presented and thoroughly analyzed.

**Questions For Authors:**

See Comments.

**Relation To Broader Scientific Literature:**

The paper is mainly related to heterogeneous Large Language Model (LLM) serving, specifically exploring how a heterogeneous GPU configuration can reduce cost budget while improving the LLM inference efficiency.

**Theoretical Claims:**

The paper did not make any theoretical claims or proofs.


##

---

> ### Author Rebuttal · Authors · 2025-04-01
>
> > 1. How the throughput h is profiled or defined?
>
> We agree with the reviewer that, because $h$ is recursively defined, it is impractical to profile every possible configuration exhaustively. However, in practice, this can be solved by employing a one-time profiling strategy that captures the following components. (This approach is referred to the profiling method used in Vidur MLSys'24.)
> - Inference prefilling latency: We profile the latency for a single transformer layer across varying tensor parallelism (TP) degrees, workload types, and GPU types.
> - Inference decoding latency: We profile the decoding latency for a single transformer layer under similar variations in TP degrees, workload types, and GPU types.
> - Pipeline communication latency: We measure the communication latency between different GPUs across various workload types.
>
> Using these measurements, the per-request latency for any configuration is estimated by combining the TP costs (both communication and computation) of all layers—which may be served by different GPUs and at varying TP degrees—with the pipeline parallelism (PP) communication cost. (Note that our heuristics, as discussed in Section 4.3 and Appendix D, largely reduce the profiling space, e.g., TP is employed only intra-machine.)
>
> Additionally, when estimating throughput, the prefill and decoding phases are treated separately: **(1)** The prefill phase is compute-bound, and its batched processing capacity is determined by the sum of the individual latencies; **(2)** the decoding phase is memory-bound, with its batched processing capability defined by a single latency value. This distinction has been validated in several studies (DistServe OSDI'24, Splitwise ISCA'24).
>
> |Diff Configs|Real (req/s)|Estimated (req/s)|
> |-|-|-|
> | H100(2,4)|0.56|0.60|
> | H100(4,2)|0.44|0.47|
> | H100(4,2)(cross machine)|0.42|0.44|
> | L40(2,4)|0.42|0.46|
> | L40(4,2)|0.21|0.22|
> | L40(4,2)(cross machine)|0.18|0.19|
> | H100+A100(4,2)(cross machine)|0.48|0.52|
>
> The table demonstrates examples of our cost estimation under long input and short output workload (i.e., workload 1 in Figure 4), the notation (2,4) indicates that the TP degree is 2 and the PP degree is 4. Although the estimations are not perfectly accurate, they are sufficiently reliable (estimation errors within 4%-7%) for selecting the optimal configurations.
>
> We will integrate the details of the one-time profiling into the updated version of our paper.
>
> > 2. Workload distribution absent from MILP formulation.
>
> Thanks for the reviewers' detailed and in-depth understanding of our paper. Our workload assignment method ensures that every workload type is completely allocated across all configurations, as enforced by the constraint: $\sum_{c \in C} x_{c,w} = 1, \forall\,w$. Although these variables are normalized to sum to 1, they can be scaled by the actual workload counts (e.g., 500 requests for type 1300 requests for type 2) during implementation.
>
> To avoid misunderstadning, we will revise the formulation in the updated version of our paper: "Let $f_w$ be the total number of requests for workload $w$. The time required on $c$ is given by $T_c = \sum_{w=1}^{W} \frac{x_{c,w} \cdot f_w}{y_c \cdot h_{c,w}}$."
>
> > 3. Integrating more heterogeneous baseline.
>
> ||Llama-30B|Llama3-70B|
> |-|-|-|
> |Helix |8.49 req/s|5.72 req/s|
> |Ours |11.43 req/s (35%↑)|7.13 req/s (25%↑)|
>
> We conduct additional experiments comparing our system with Helix (ASPLOS'25). Specifically, we compare our method against Helix under a price budget of $15 per hour on the AzureTrace dataset. While Helix optimizes heterogeneous LLM serving using max-flow and MILP scheduling, our method explicitly considers workload heterogeneity and GPU composition optimization, resulting in better cost efficiency. Experimental results show that our system outperforms Helix by 25–35%.
>
> We will integrate the baseline into our updated paper.
>
> > 4. Additional Suggestions.
>
> Thanks for the reviewers' detailed review. We will solve the following mistakes and update our paper:
> - We will update Figure 3, 4, 11, 12, 13 using "Throughput Per Dollar".
> - We will explain that the x sticks represent P5 Latency, P10 Latency, P15 Latency, etc.
> -
>     - We will update Figure 4, 12, 13 using "Different Deployment Configurations" instead of "Different GPU types".
>     - We will clarify that different bars within Figure 4 represent different deployment methods, and the optimal method varies according to workload, model type, and GPU type.
> - We will correct m and M to w and W.
> - We will check all the cited papers and update the references to the published ones, e.g., sarathi-serve (OSDI'24), HexGen (ICML'24), WildChat (ICLR'24) etc.

---

> > ### Comment · Reviewer_2VxX · 2025-04-02
> >
> > Thanks for your efforts and clarification. The rebuttal has successfully addressed some of my concerns, and I have updated my score accordingly.

---

> > > ### Author Response · Authors · 2025-04-02
> > >
> > > Thank you for your acknowledgment. We will revise the paper according to your suggestion.

---

### Official Review · Reviewer_79GJ · 2025-03-12

**Overall Recommendation:** 3

**Summary:**

This paper investigates cost-efficient LLM serving on heterogeneous GPUs. It benchmarks various GPU types and develops a MILP-based scheduling algorithm to optimize deployment configurations. The study shows that leveraging heterogeneous GPUs improves cost efficiency compared to homogeneous setups.

**Claims And Evidence:**

Claim: Heterogeneous GPU compositions improve cost-efficiency over homogeneous deployments.
Evidence: Benchmarking results demonstrate that different GPU types align well with varying workload demands, leading to improvement in cost-efficiency.

Claim: Optimal deployment configurations significantly affect performance.
Evidence: Experimental results show that tuning deployment strategies (e.g., tensor parallelism vs. data parallelism) improves performance.

Claim: The proposed MILP-based scheduling algorithm outperforms existing heterogeneous serving frameworks.
Evidence: Comparisons with HexGen show up to 18% improvement in throughput.

**Essential References Not Discussed:**

QLM (SoCC 2024): Addresses SLO-aware LLM serving and multi-node optimizations.
Sarathi Serve (OSDI 2024): Implements prefill chunking for hybrid execution batches.
Vidur (MLSys 2024): Solves the problem of deployment tuning

**Experimental Designs Or Analyses:**

While the evaluation is thorough, there are several weaknesses remain:
The paper does not address how the system handles sudden bursts of traffic or fluctuating real-time workloads.
While the proposed method improves cost-efficiency, it lacks details on handling conflicts between prioritizing throughput and latency.

The paper does not clearly articulate why adding budget constraints to existing LP solvers is challenging, given prior work like Melange.

**Methods And Evaluation Criteria:**

The study evaluates its approach on real-world and synthetic workload traces, including Llama3-8B and Llama3-70B models. The study lacks a discussion on the impact of fluctuating real-time workloads.

**Other Comments Or Suggestions:**

How does the approach handle workload spikes and dynamic GPU availability fluctuations?
What are the trade-offs between prioritizing cost-efficiency and request latency?
Under what scenarios does the MILP-based scheduling significantly diverge from existing heterogeneous scheduling frameworks?

**Other Strengths And Weaknesses:**

see rest of the review.

**Questions For Authors:**

Please see above

**Relation To Broader Scientific Literature:**

This is a growing area but novelty with respect to existing literature is think or not clear.

**Theoretical Claims:**

The paper does not introduce new theoretical models but formulates GPU scheduling as a MILP problem.

---

> ### Author Rebuttal · Authors · 2025-04-01
>
> > 1. Workload spikes and dynamic GPU availability fluctuations.
>
> Online rescheduling to adapt to workload changes and GPU drops is an interesting idea that can easily be integrated into our current solution. We introduce this approach and present some preliminary experimental results.
>
> **Solution: Online replanning.** To address the reviewer's concern, we implement an online replanning mechanism analogous to the one proposed in DistServe (OSDI'24). Concretely, the system could **(1)** monitor the real-time composition of incoming workloads (mentioned in Section 3), **(2)** track GPU resource availability within the cloud environment, and **(3)** upon detecting a significant shift in workload distribution, (e.g., an increase in the proportion of certain workload types) the scheduling algorithm could be executed again, incorporating recent historical data to produce an updated serving plan.
>
> | Status| Workload Change| GPU Drop|
> |------------|-------------------------|--------------------|
> | Before| 26.89 req/s| 26.89 req/s|
> | After| 23.70 req/s (13%↓)| 20.80 req/s (29%↓) |
> | Replanning| 29.61 req/s (25%↑)| 22.85 req/s (10%↑)|
>
> **Replanning results.** **(1)** We test the workload surge in short output requests in AzureTrace. Before surge, the optimal GPU composition is {20%, 65%, 15%} for datacenter, workstation, and consumer GPUs, achieving 26.89 req/s. After workload change, the throughput degrades to 23.7 req/s. In this case, replanning (shifting allocation to {63%, 23%, 14%}) boosts throughput to 29.61 req/s. **(2)** We also test the case when GPU drop happens (4 H100s down), the throughput falls from 26.89 to 20.80 req/s. In this case, replanning raises throughput to 22.85 req/s. Additionally, the rescheduling and model-weight reloading phases can be completed within 1–2 minutes, which is significantly shorter than the hourly timescale of real-world workload changes.
>
> We will integrate this discussion into Section 6 of our updated draft.
>
> > 2. The trade-offs between prioritizing cost-efficiency and request latency.
>
> **Trade-offs.** We acknowledge that there are trade-offs between optimizing cost-efficiency and latency. **(1)** Prioritizing cost-efficiency typically involves using fewer resources (i.e., lower budgets), which can lead to slightly higher response latencies. **(2)** In contrast, prioritizing latency often requires utilizing more resources (i.e., incurring higher costs).
>
> **Our focus.** Despite the trade-offs, our work mainly focuses on the cost-efficiency for two reasons. **(1)** Some inference tasks do not require extremely low latency; meeting a predefined latency threshold (e.g., reading/speaking speed) is usually sufficient. And more importantly, **(2)** in resource-limited scenarios, where systems are naturally under-provisioned, emphasizing throughput can also indirectly improve latency by reducing queuing delays. Our experimental results in Figure 6 demonstrate that our method achieved the lowest P99 latency among all compared baselines.
>
> We will integrate the above discussion into Section 6 of the updated draft.
>
> > 3. Why adding budget constraints to existing LP solvers (e.g., Melange) is challenging?
>
> Existing heterogeneous frameworks require substantial redesign of their scheduling algorithms or significant additional system development to achieve cost optimization comparable to our approach.
>
> **Compare with Melange.** **(1)** Melange assigning each workload to a *single* GPU type, overlooking the impact of parallelism strategies on resource allocation. As shown in Section 3, tuning parallelism strategies is crucial, and incorporating this dimension significantly expands the search space and demands additional system development. **(2)** Melange assumes unlimited resources, yet in practice, GPU availability and budget constraints are real issues that necessitate evaluating workloads across *multiple* GPU types to find a globally optimal solution—further expanding the search space and rendering Melange’s approach impractical.
>
> **Compare with HexGen and Helix.** Both approaches **(1)** fail to consider workload heterogeneity and the presence of mixed workloads during scheduling, and **(2)** their scheduling algorithms are designed for fixed heterogeneous cluster configurations, missing opportunities for further performance improvements achievable through GPU composition optimization.
>
> > 4. Essential References Not Discussed.
>
> QLM (SoCC 2024) focuses on SLO-aware serving and multi-node optimizations by refining request ordering; Sarathi Serve (OSDI 2024) optimizes batching through prefill chunking to mitigate interference between the prefill and decoding stages; and Vidur (MLSys 2024) develops an accurate simulator for deployment tuning.
>
> Our work has a different goal—it is dedicated to achieving heterogeneous, cost-efficient serving in cloud environments. We will integrate the discussion of these references into Section 2 of the updated draft.

---

### Official Review · Reviewer_nHPD · 2025-03-14

**Overall Recommendation:** 3

**Summary:**

This paper focuses on the cost efficiency of LLM services on heterogeneous GPUs, proposing ways to improve efficiency by optimizing GPU composition, deployment configurations, and workload allocation.

**Claims And Evidence:**

When discussing related work, the paper mainly emphasizes that other methods do not consider GPU heterogeneity and user budget constraints, but does not analyze in depth the specific shortcomings of existing methods, e.g., whether pre-existing methods can achieve similar cost-optimization results by adjusting their strategies.

**Essential References Not Discussed:**

No

**Experimental Designs Or Analyses:**

The paper's experiments are mainly based on the Vast.ai platform, which has small GPU availability and the experimental results may be difficult to generalize to large-scale cloud environments.

**Methods And Evaluation Criteria:**

The paper assumes that task assignments are all computed before the task is executed, while in real cloud environments, task arrivals are dynamic and the paper does not provide an online scheduling strategy that adapts to dynamic changes in tasks.

**Other Comments Or Suggestions:**

No

**Other Strengths And Weaknesses:**

Strengths:
The organization of this paper is good.

Weaknesses:
1. The research problem of the paper focuses too much on the cost-effective optimization of heterogeneous GPUs, but other factors that may affect the cost (e.g., network latency, energy consumption, computational complexity of scheduling algorithms, etc.) are under-explored.

2. The paper fails to address more complex scheduling challenges in cloud environments, such as scheduling GPU resources across data centers and uncertainty due to preemptive instances.

**Questions For Authors:**

No

**Relation To Broader Scientific Literature:**

The paper only compares a few heterogeneous LLM scheduling methods and lacks comparative analysis with a wider range of scheduling optimization methods (e.g., reinforcement learning methods, heuristic scheduling methods)

**Theoretical Claims:**

Yes

---

> ### Author Rebuttal · Authors · 2025-04-01
>
> > 1. Specific shortcomings of existing methods.
>
> Existing methods require a heavy redesign of the scheduling algorithms or demand significant additional system development to achieve similar cost optimization.
>
> **Compare with HexGen and Helix.** Both approaches (1) fail to consider workload heterogeneity and the presence of mixed workloads during scheduling, and (2) their scheduling algorithms are designed for fixed heterogeneous cluster configurations, missing opportunities for GPU composition optimization.
>
> **Compare with Melange.** Melange does not consider the impact of parallelism strategies or cloud resource constraints on deployment performance. Its method fails to utilize different parallelism strategies or different GPU types to handle various workloads.
>
> > 2. Adapting to dynamic tasks and preemption uncertainty.
>
> Online scheduling to adapt to workload changes and GPU drops is an interesting idea that can easily be integrated into our current solution:
>
> **Online replanning.** We implement an online replanning mechanism analogous to the one proposed in DistServe (OSDI'24). Concretely, the system could (1) monitor the real-time composition of incoming workloads, (2) track GPU resource availability within the cloud environment, and (3) upon detecting a significant shift in workload distribution, the scheduling algorithm could be executed again, incorporating recent historical data to produce an updated serving plan.
>
> || Workload Change|GPU Drop|
> |-|-|-|
> |Before|26.89(req/s)|26.89|
> |After|23.70(13%↓)|20.80(29%↓)|
> |Replanning|29.61(25%↑)|22.85(10%↑)|
>
> **Experimental Results.** (1) We test the workload surge in short output requests in AzureTrace. Before surge, the optimal GPU composition is {20%, 65%, 15%} for datacenter, workstation, and consumer GPUs, achieving 26.89 req/s. After workload change, the throughput degrades to 23.7 req/s. In this case, replanning (shifting allocation to {63%, 23%, 14%}) boosts throughput to 29.61 req/s. (2) We also test the case when GPU drop happens (4 H100s down), the throughput falls from 26.89 to 20.80 req/s. In this case, replanning raises throughput to 22.85 req/s. Additionally, the rescheduling and model-weight reloading phases can be completed within 1–2 minutes, which is significantly shorter than the hourly timescale of real-world workload changes.
>
> > 3. Vast.ai platform has small GPU availability.
>
> GPU shortages are a well-documented challenge in cloud-based tasks (SkyPilot NSDI'23). This issue is not unique to Vast.ai—similar GPU availability limitations are observed across many widely-used platforms, including FluidStack, DataCrunch, and RunPod. Even major cloud providers, such as Google Cloud, face GPU quota constraints, as noted in Table 4 of Helix (ASPLOS'25), with quotas being limited and varying across regions.
>
> > 4. Compare other scheduling methods.
>
> **Scheduling algorithm comparison.** We further compare our method with a heuristic scheduling method by replacing the plan optimization component of MILP with population-based mutation and selection (i.e., genetic algorithm). In a 48-GPU cluster, the heuristic method requires 115 seconds to achieve comparable performance to our MILP formulation, which only requires 30 seconds.
>
> **End-to-end system comparison.** We compared our method with HexGen in our paper, here, we further compare our system with Helix (ASPLOS'25), which optimizes heterogeneous LLM serving using max-flow and MILP scheduling, under a price budget of $15 per hour on the AzureTrace dataset. Our method explicitly considers workload heterogeneity and GPU composition optimization, resulting in greater cost efficiency.
>
> ||Llama-30B|Llama3-70B|
> |-|-|-|
> |Helix|8.49(req/s)|5.72|
> |Ours|11.43(35%↑)|7.13(25%↑)|
>
> > 5. Other factors that may affect the cost.
>
> **Network Latency impact.** We have incorporated network latency considerations into our MILP formulation. For instance, (1) the GPU-to-GPU communication latency is used to determine the pipeline communication cost, and (2) tensor parallelism with high communication volumes is prioritized to use intra-machine networks.
>
> **Energy consumption.** Our focus is optimizing the cost efficiency of cloud-based LLM serving; optimizing energy consumption is orthogonal to our paper’s scope. However, we acknowledge that energy consumption is a critical metric and could be considered as a future direction.
>
> **Computational complexity.** Thank you for mentioning this. We will include a detailed discussion of computational complexity of the MILP formulation in Section 4.3 of the updated draft: “the theoretical worst-case time complexity scales as: O(poly(|C|,W,N)×2^|C|), the polynomial factor accounts for the overhead of processing each node in the search tree”.
>
> > 6. Scheduling across data centers.
>
> We have already used GPUs across data centers in our heterogeneous experiment setup, since Vast.ai provides different GPUs in different data centers (e.g., A40 and A100 GPUs reside in Australia and New Jersey, US).

---

### Decision · Program_Chairs · 2025-05-01

**Decision:**

Accept (poster)

**Comment:**

The following issues have been pointed out in this paper:
- Lack of consideration for dynamic workloads and preemptible GPU instances
- Limited baseline comparisons
- Theoretical and modeling assumptions
- Trade-off between latency and cost-efficiency
- Presentation issues and missing references

However, the following points are evaluated positively, and based on an overall assessment, the recommendation is Weak Accept:
- Timely and practically relevant problem (LLM serving cost-efficiency on heterogeneous GPUs)
- Solid MILP-based optimization formulation addressing real-world deployment constraints
- Extensive and realistic experiments using modern LLMs and real workload traces
- Introduction of an online replanning mechanism for dynamic adaptation

That said, neither the reviewers nor the Area Chair strongly recommend acceptance of this paper.